# Reconstructing Perceptive Images from Brain Activity by Shape-Semantic GAN

**Tao Fang**[1], **Yu Qi**[1,*], **Gang Pan**[2,1,3*]

duolafang@zju.edu.cn, qiyu@zju.edu.cn, gpan@zju.edu.cn

[1] College of Computer Science and Technology, Zhejiang University
[2] State Key Lab of CAD&CG, Zhejiang University
[3] The First Affiliated Hospital, College of Medicine, Zhejiang University

## Abstract

Reconstructing seeing images from fMRI recordings is an absorbing research area in neuroscience and provides a potential brain-reading technology. The challenge lies in that visual encoding in brain is highly complex and not fully revealed. Inspired by the theory that visual features are hierarchically represented in cortex, we propose to break the complex visual signals into multi-level components and decode each component separately. Specifically, we decode shape and semantic representations from the lower and higher visual cortex respectively, and merge the shape and semantic information to images by a generative adversarial network (Shape-Semantic GAN). This 'divide and conquer' strategy captures visual information more accurately. Experiments demonstrate that Shape-Semantic GAN improves the reconstruction similarity and image quality, and achieves the state-of-the-art image reconstruction performance.

## 1 Introduction

Decoding visual information and reconstructing stimulus images from brain activities is a meaningful and attractive task in neural decoding. The fMRI signals, which record the variations in blood oxygen level dependent (BOLD), can reveal the correlation between brain activities and different visual stimuli by monitoring blood oxygen content. Implementing an fMRI-based image reconstruction method can help us understand the visual mechanisms of brain and provide a way to 'read mind'.

**Previous studies.** According to previous studies, the mapping between activities in visual cortex and visual stimulus is supposed to exist [19], and the perceived images are proved to be decodable from fMRI recordings [29, 16, 6]. Early approaches estimate the mapping using linear models such as linear regression [16, 6, 7, 17, 18]. These approaches usually firstly extract specific features from the images, for instance multi-scale local image bases [16] and features of gabor filters [29], then learn a linear mapping from fMRI signals to image features. Linear methods mostly focus on reconstructing low-level features, which is insufficient for reconstructing complex images, such as natural images. After the homogeneity between the hierarchical representations of the brain and deep neural networks (DNNs) was revealed [9], methods based on this finding have achieved great reconstruction performance [23, 28]. Shen *et al.*[23] used convolutional neural networks (CNN) models to extract image features and learned the mapping from fMRI signals to the CNN-based image features, which successfully reconstructed natural images. Recently, the development of DNN makes it possible to learn nonlinear mappings from brain signals to stimulus images in an end-to-end manner [24, 1, 30, 14, 3]. The DNN-based approaches have remarkably improved the reconstruction performance, such as Encoder-Decoder models [1, 26] and Generative Adversarial Network (GAN)

based models [24, 1, 30, 14, 3]. Shen et al. *et al.* [24] proposed a DNN-based decoder which learned nonlinear mappings from fMRI signals to the seeing images effectively. Beliy et al. *et al.* [1] learned a bidirectional mapping between fMRI signals and stimulus images using an unsupervised GAN. These recent approaches achieved higher image quality and can reconstruct more natural-looking images compared with linear methods.

Learning a mapping from fMRI recordings to the corresponding stimulus images is a challenging problem. The difficulty mostly lies in that brain activity in the visual cortex is complex and not fully revealed. Studies have shown that there exists a hierarchical increase in the complexity of representations in visual cortex[5], and study [17] has demonstrated that exploiting information from different visual areas can help improve the reconstruction performance. Simple decoding models without considering the hierarchical information may be insufficient for accurate reconstruction.

The hierarchical structure of information encoding in visual areas have been widely studied [8, 2, 5]. On the one hand, activities in the early visual areas show high response to low-level image features like shapes and orientations [10, 20, 25, 15]. On the other hand, anterior visual areas are mostly involved in high-level information processing, and activities in such visual areas show high correlation with the semantic content of stimulus images [9, 5]. Such high-level image features are more categorical and invariant than low-level features in identification or reconstruction [2]. The hierarchical processing in the visual cortex inspired us to decode the low-level and high-level image features from lower visual cortex (LVC) and higher visual cortex (HVC) separately [9].

In this study, we propose a novel method to realize image reconstruction from fMRI signals by decomposing the decoding task into hierarchical subtasks: shape/semantic decoding in lower/higher visual cortex respectively (Figure 1). In shape decoding, we propose a linear model to predict the outline of the core object from the fMRI signals of lower visual cortex. In semantic decoding, we propose to learn effective features with a DNN model to represent high-level information from higher visual cortex activities. Finally, the shape and semantic features are combined as the input to a GAN to generate natural-looking images with the shape and semantic conditions. Data augmentation is employed to supplement the limited fMRI data and improve the reconstruction quality.

Experiments are conducted to evaluate the image reconstruction performance of our method in comparison with the state-of-the-art approaches. Results show that the Shape-Semantic GAN model outperforms the leading methods. The main contributions of this work can be summarized as follows:

- Instead of directly using end-to-end models to predict seeing images from fMRI signals, we propose to break the complex visual signals into multi-level components and decode each component separately. This 'divide and conquer' approach can extract visual information accurately.

- We propose a linear model based shape decoder and a DNN based semantic decoder, which are capable of decoding shape and semantic information from the lower and higher visual cortex respectively.

- We propose a GAN model to merge the decoded shape and semantic information to images, which can generate natural-looking images given shape and semantic conditions. The performance of GAN-based image generation can be further improved by data augmentation technique.

## 2    Methods

The proposed framework is composed of three key components: a shape decoder, a semantic decoder and a GAN image generator. The framework of our approach is illustrated in Figure 1. Let $x$ denote the fMRI recordings and $y$ be the corresponding images perceived by the subjects during experiments. Our purpose is to reconstruct each subject's perceived images $y$ from the corresponding fMRI data $x$. In our method, we use the shape decoder $C$ to reconstruct the stimulus images' shapes $r_{sp}$ and the semantic decoder $S$ to extract semantic features $r_{sm}$ from $x$. The image generator $G$ implemented by GAN is introduced to reconstruct the stimulus images $G(r_{sp}, r_{sm})$ in the final stage of decoding.

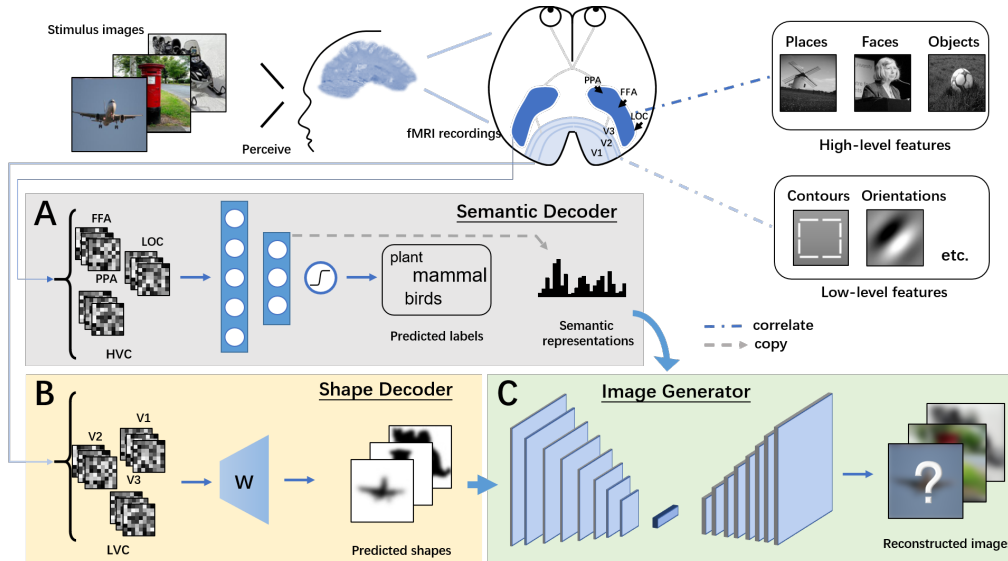

Figure 1: The framework of the proposed method. The decoding task is divided into two parts: training (A) a semantic decoder (extracting high-level features) and (B) a shape decoder (extracting low-level features) to decode from higher and lower visual cortex respectively. The decoded shape and semantic information are input to (C) a GAN model to reconstruct the seeing images.

## 2.1 Dataset

We make use of a publicly available benchmark dataset from [23]. In this dataset brain activity data were collected in the form of functional images covering the whole brain. The corresponding stimulus images were selected from ImageNet including 1200 training images and 50 test images from 150 and 50 categories separately. Images in both of the training and test set have no overlap with each other. And each image has 5 or 24 fMRI recordings for training or test respectively.

The fMRI signals contain information from different visual areas. Early visual areas like V1,V2 and V3 were defined by the standard retinotopic mapping procedures [4, 22, 23], and V1,V2 and V3 were concatenated as an area named lower visual cortex [9]. Higher visual cortex is composed of regions including parahippocampal place area (PPA), lateral occipital complex (LOC) and fusiform face area (FFA) defined in [9].

## 2.2 Shape Decoder

In order to obtain the outline of the visual stimulus, we present a shape decoder $C$ to extract low-level image cues from the lower visual cortex based on linear models. Using a simple model to obtain low-level visual features, which has been demonstrated feasible by previous studies [29], can avoid the overfitting risk of complex models. Shape decoder $C$ consists of three base decoders trained for V1, V2, V3 individually and a combiner to merge the results of base shape decoders. The process of shape decoding is described in Figure 2.

The stimulus images are firstly preprocessed by shape detection and feature extraction before shape decoder training.

**Shape detection.** First, image matting is conducted on the stimulus images to extract the core objects [13] and remove the interference of other parts in the images. The objects in the stimulus images are extracted based on saliency detection [12] and manual annotation. The results are binarized to eliminate the influence of minor variance in images and emphasize objects over background. By such preprocessing method, the core objects are extracted and some details (like colors or textures) that help little in shape decoding are eliminated.

**Features extraction.** Second, square image patches are presented for feature extraction. Pixel values located in non-overlapping $m \times m$ pixels square patches are averaged as the image patch's value. By

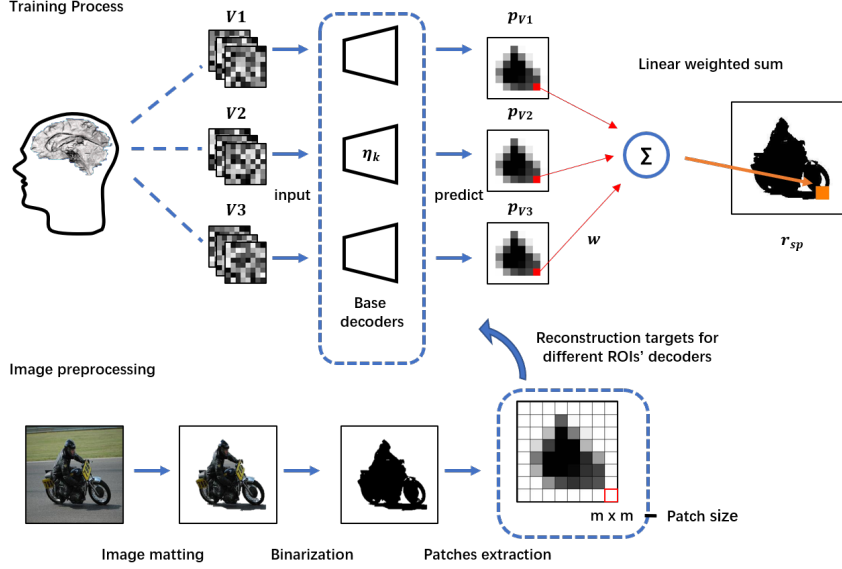

Figure 2: Flow chart of the shape decoder. The linear models are trained to predict the shapes from V1 to V3 individually, and then the intermediate results $p_k$ are combined to get the decoded shapes $r_{sp}$.

representing shapes in contrast-defined patch images, the amount of calculation can be reduced and improve the invariance to small distortions. In our model $m = 8$ is selected.

**Model Training.** Because V1, V2 and V3 areas have representations for the visual space respectively, we train the base decoders for each of them and use a linear weighting combiner to combine the decoded shapes. The values of the image patches are normalized to [0, 1] and flattened to one-dimensional vectors $p$. The decoded shape vectors $p_k^* = c_k(x_k)$, where $c_k(x_k)$ is the base shape decoder whose parameter $\eta_k$ is optimized by:

$$\eta_k = \arg\min_{\eta_k} \ \|c_k(x_k) - p_k\|, \quad k = V1, V2, V3, \tag{1}$$

where $\eta_k$ denotes the weights of base shape decoder $c_k$ and $k$ denotes the visual area that the samples belong to. The base decoder $c_k$ is implemented by linear regression and the models are trained for fMRI recordings in V1, V2 and V3 individually. Then a combiner is trained to combine the predicted results $p_{V1}^*$, $p_{V2}^*$ and $p_{V3}^*$:

$$r_{sp}(i,j) = \sum_k w_{ij}^k p_k^*(i,j), \quad k = V1, V2, V3, \tag{2}$$

where $r_{sp}$ refers to the predicted shapes computed by the combiner, and $r_{sp}(i,j)$ is the pixel value at position $(i,j)$. The results $r_{sp}$ predicted by the combiner are resized to the same size as the stimulus images ($256 \times 256$ pixels). $w_{ij}$ is the weight of the combiner for pixel at position $(i,j)$. The combining weight $w_{ij}$ is computed independently for each pixel.

## 2.3 Semantic Decoder

To render semantically meaningful details on shapes, a semantic decoder is used to provide categorical information. Although images can be rendered only based on shapes with a pre-trained GAN model, in practice we find that the results are not always acceptable because of the lack of conditions. The mapping from shapes to real images is not unique in many cases (e.g. a circular shape can be translated into a football/crystal ball/golf ball and all of these translations are correct judged by the discriminator). Besides, noise retained in shapes will interfere with the reconstruction quality in the absence of other conditions. Therefore reconstructing only on the shape condition is not sufficient. A

semantic context, which is used to guide the GAN model with the image's category, can be helpful when incorporated with the shape features in training phase.

The input to the semantic decoder is fMRI signals in HVC. HVC covers the regions of LOC, FFA and PPA, whose voxels show significantly high response to the high-level features such as objects, faces or scenes respectively [9]. As showed in Figure 1, a lightweight DNN model is introduced to generate semantic features. The DNN model consists of one input layer (the same size as the input's number of voxels), two hidden layers and one output layer. Tanh activation function is introduced between the hidden layers and sigmoid activation function is used for classification. When training the DNN model, the fMRI recordings in HVC are identified by the model to infer their corresponding stimulus images' categories. After the training phase, the DNN model works as a semantic decoder. Note that the penultimate layer of DNN performs as a semantic space supporting the classification task at the output layer [28], the features in such layer are adopted as the semantic representation of fMRI signals in our method.

## 2.4   Image Generator

To reconstruct images looking more realistic and filled with meaningful details, an encoder-decoder GAN, referring to the image translation methods [11], is introduced in the final stage of image reconstruction.

In image reconstruction, there exists lots of low-level features (like contours) shared between the input shapes and the output natural images, which need to be passed across the decoder directly to reconstruct images with accurate shapes. Therefore, we propose the U-Net [21], an encoder-decoder structure, with skip connections. Traditional encoder-decoder model passes information through a bottleneck structure to extract high-level features, while low-level features such as shapes and textures can be lost. And few shape features retained in the output can cause deformation in reconstruction. By using the U-Net structure, more low-level features can be passed from the input space to the reconstruction space with the help of skip connections without the limitation of the bottleneck.

The generator is composed of a pair of symmetrical encoder and decoder. The encoder and decoder have eight convolutional or deconvolutional layers with symmetrical parameters and no down-sampling or up-sampling is used. The input layer takes the $256 \times 256$ pixel images (shape images) as input. The bottleneck between encoder and decoder represents the high-level features extracted by convolutional layers in encoder, which is modified to take both of the semantic features $r_{sm}$ and the high-level features as input to the decoder. In this way the generator will be optimized under the constraint of semantic and shape conditions. The discriminator takes the shape $r_{sp}$ and the output of generator together as input and predicts the similarity of high-frequency structures between these two domains, using this similarity to guide the generator training.

Let $G_\theta$ denotes the U-Net generator and $D_\phi$ denotes the discriminator. The generator $G_\theta$ and discriminator $D_\phi$ have parameters named $\theta$ and $\phi$, which are optimized by minimizing the loss function $L(\theta, \phi)$. The objective of the conditional GAN is composed of two components, which can be described as

$$L(\theta, \phi) = L_{adv}(\theta, \phi) + \lambda_{img} L_{img}(\theta), \tag{3}$$

where $L_{adv}(\theta, \phi)$ and $L_{img}(\theta)$ denote the adversarial loss and image space loss, and $\lambda_{img}$ define the weight of the image space loss $L_{img}$ in $L(\theta, \phi)$. As is inferred in [11], L1 loss is able to accurately capture the low frequencies. The GAN discriminator is designed to model the high-frequency structures. By combining these two terms in the loss function, blurred reconstructions will not be tolerated by the discriminator and low-frequency visual features can also be retained at the same time.

The adversarial loss and the image space loss used in optimizing the generator can be expressed as

$$L_{adv}(\theta, \phi) = -E_{r_{sp}, r_{sm}}[log(D_\phi(r_{sp}, G_\theta(r_{sp}, r_{sm})))], \tag{4}$$

$$L_{img}(\theta) = E_{r_{sp}, r_{sm}, y}[\|y - G_\theta(r_{sp}, r_{sm})\|], \tag{5}$$

where $r_{sp}$, $r_{sm}$ and $y$ refer to shapes, semantic features and stimulus images. During the training phase, gradient descent is computed on $G_\theta$ and $D_\phi$ alternately. Instead of directly training $G_\theta$ to minimize $log(1 - D_\phi(r_{sp}, G_\theta(r_{sp}, r_{sm})))$, we followed the recommendations in [24] and maximize $log(D_\phi(r_{sp}, G_\theta(r_{sp}, r_{sm})))$. The objective of the discriminator is:

$$L_{discr}(\theta, \phi) = -E_{r_{sp},y}[logD_\phi(r_{sp}, y)] - E_{r_{sp},r_{sm}}[log(1 - D_\phi(r_{sp}, G_\theta(r_{sp}, r_{sm})))]. \quad (6)$$

When $G_\theta$ is being trained, it tries to optimize $\theta$ to reduce the distance between generated images $G_\theta(r_{sp}, r_{sm})$ and stimulus images $y$. It also tries to generate images that share similar high-frequency structure with shapes $r_{sp}$ to confuse $D_\phi$ and let $D_\phi$ predict $G_\theta(r_{sp}, r_{sm})$ as correct. When $D_\phi$ is trained, it tries to optimize $\phi$ to distinguish the pairs of $\{r_{sp}, y\}$ from the pairs of $\{r_{sp}, G_\theta(r_{sp}, r_{sm})\}$. Each time one of $G_\theta$ or $D_\phi$ is trained, the other's parameters are fixed.

## 2.5 Data Augmentation

Since the size of the fMRI dataset is limited, we propose to improve the image reconstruction performance by data augmentation in GAN training.

We sampled the augmented images from the ImageNet dataset. For shape augmentation, the preprocess in Section 2.2 is conducted on augmented images and the contrast-defined, $m \times m$-patch images $R_{sp}$ represent as shapes of the augmented images. For semantics augmentation, the category-average semantic feature $R_{sm}$ is computed as a substitute of semantic vector. $R_{sm}$ is defined as the vector obtained by averaging the semantic features of samples annotated with the same category. By combining the shapes and category-average semantic features generated from the augmented images as the form of $\{R_{sp}, R_{sm}\}$ pairs, the new samples are concatenated with the $\{r_{sp}, r_{sm}\}$ pairs as the inputs to $G_\theta$, enhancing the generality of $G_\theta$ eventually. Note that in our method the image augmentation could only be conducted within images that corresponds to the same classes as the training images. In reconstruction about 1.2k augmented natural images are randomly selected from the same image dataset as [23] (ILSVRC2012), and they have no overlap with the training or test set.

## 2.6 Implementation Details

We implemented the image generator using the PyTorch framework and modified the image translation model provided by [11]. The image generator consists of a U-Net generator $G$ and a discriminator $D$. In both of $G$ and $D$, the kernel size is (4, 4), step size is (2,2) and the padding size is 1 for parameters of the layers. The generator is composed of 8 parametrically symmetric convolutional/deconvolutional layers with LeakyReLU (0.2) used as activation functions. All the input images (the stimulus images and shape images) of $G$ and $D$ are resized to (256, 256, 3).

In GAN training, minibatch SGD is used and Adam solver is employed to optimize the parameters with momentum $\beta_1 = 0.9$ and $\beta_2 = 0.999$. The initial learning rate is $2 \times 10^{-4}$ and 10 samples are input in a batch. The weights of individual loss terms affect the quality of the reconstructed image. In our experiments, we set $\lambda_{img} = 100$ to make a balance between the results' sharpness and similarity with stimulus images. The image generator is trained for 200 epoch totally with the learning rate decay occurring at 120 epoch.

# 3 Results

To evaluate the quality of reconstructed images, we conduct both visual comparison and quantitative comparison. In quantitative comparison, pairwise similarity comparison analysis is used to measure the reconstructed images' quality, which is introduced in [24]. One reconstructed image is compared with two candidate images (the ground-truth image and a randomly selected test image) to test if its correlation with the ground truth image is higher. And in our experiments structural similarity index (SSIM) [27] is used as the correlation measure. SSIM measures the similarity of the local structure between the reconstructed and origin images in spatially close pixels [24].

## 3.1 Comparison of Image Reconstruction Performance

Here we compare the image reconstruction performance with existing approaches. The competitors include [1], [24], and [23]. For visual comparison, we directly use the reconstructed images reported in the papers of [1], [24], and [23], respectively. For quantitative comparison, we use the reported pairwise similarity with SSIM for [24]. For [1], we run the code published along with the paper, and use the same data augmentation images as our approach. All the pairwise similarity results are averaged by five runs to mitigate the effectiveness of randomness.

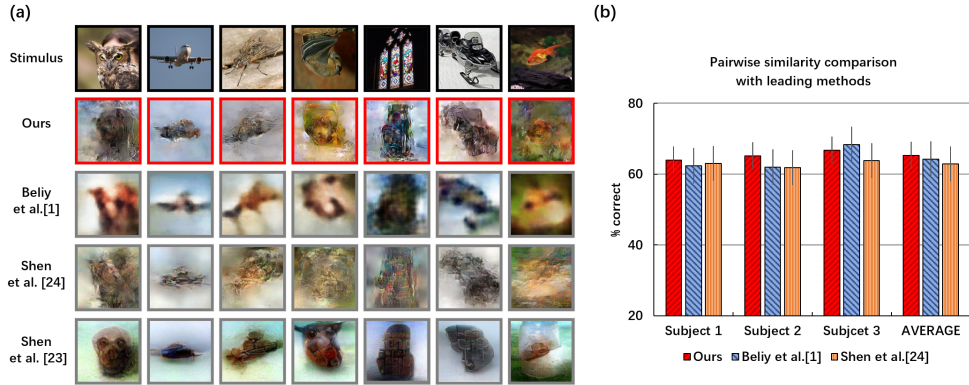

Figure 3: Image reconstruction performance comparison with other methods. (a) Images reconstructed by different methods. (b) Performance comparison with pairwise similarity.

Samples of reconstructed images are presented in Figure 3, in comparison with existing approaches. Similar to [23], the test fMRI samples corresponding to the same category are averaged across trials to improve the fMRI signals' signal-to-noise ratio (SNR). The results are reconstructed from the test-fMRI recordings of three subjects (150 samples totally), and the performance of this model is compared with the leading methods on the same dataset [23]. In visual comparison, we compare our reconstructed images with methods in [23, 24] and [1] in Figure 3a. Owing to the U-Net model trained with semantic information, our model's reconstructed images are vivid and close to the real stimulus images in color. Also, under the constraint of shape conditions, the reconstructed images share similar structures with the origin images. In quantitative comparison, we conduct pairwise similarity comparison based on SSIM with the existing methods of [1] and [24]. The comparison of different approaches used on three subjects' fMRI recordings are displayed in Figure 3b. Results show that our method performs slightly better than [1] (ours 65.3 % vs. 64.3 % on average) and outperforms [24] (62.9% on average).

## 3.2    Decoding Performance of Different ROIs

In this experiment, we evaluate the decoding performance of shape/semantic information from different ROIs (region of interest). Forty samples in the origin training set are reserved for validation and the rest are used for the decoders' training in this experiment.

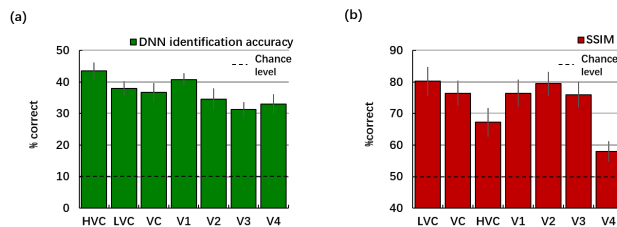

Figure 4: Decoding performance of different ROIs. (a) Performance of semantic decoding with different ROIs. (b) Performance of shape decoding with different ROIs.

To compare different ROIs' semantic representation performance, the semantic decoders (DNN models) are trained on fMRI signals in different visual areas individually. The trained DNN models are used to decode semantic features from the validation fMRI samples and identify their corresponding categories. To facilitate the comparison, we use 10-category rough labels in this section (see supplementary materials). The identification accuracy of semantic representations decoded from different ROIs is compared in Figure 4a. Results show that semantic representations extracted from the fMRI data in HVC outperform those extracted from other areas such as LVC (by 14.5%), suggesting that voxels in more anterior areas like HVC show high correlation with abstract features.

To compare the decoding performance of shape features from different ROIs, we train shape decoders from different visual areas respectively. The similarity between the decoded shapes and the stimulus images' shapes is measured by the pairwise similarity comparison based on SSIM in Figure 4b. Results show that decoding shapes from fMRI data in LVC perform better than in other areas like HVC (by 19.3%), indicating that signals in LVC have high response to low-level image features and details.

The findings that improved performance can be achieved when different decoding models are trained for low/high-level features with lower/higher visual areas respectively, are also in line with previous studies [17, 9]. In our experiments, models trained on the whole visual cortex (VC) perform slightly worse than those only trained on LVC/HVC in shape/semantic decoding tasks, probably because of the interference caused by low-correlation visual areas in VC (such as introducing higher visual areas in decoding low-level features like shapes). Note that theoretically the information processed in the HVC should also be contained in LVC in the form of low level features, it can be inferred that semantic decoding from signals in LVC may perform better when using a deeper model.

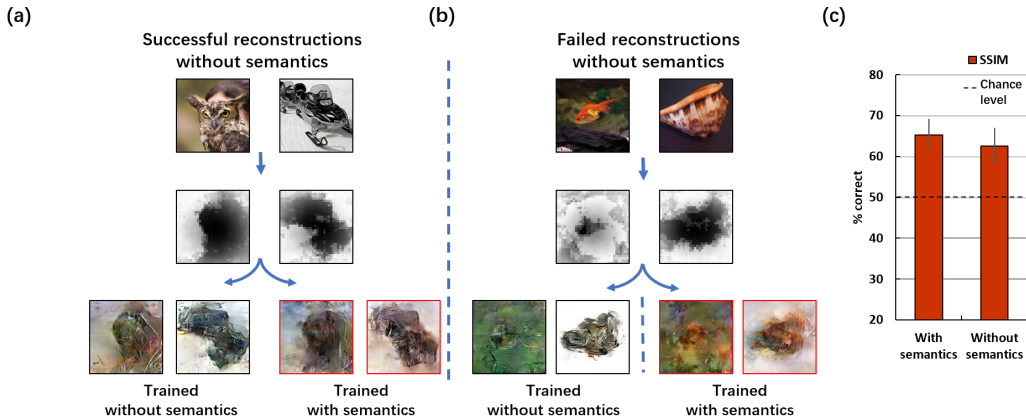

Figure 5: Effectiveness of semantics. (a) Training without semantics works well on part of the samples. (b) Failed/successful reconstructions on some samples without/with semantic conditions. (c) Quantitative comparison of reconstruction with/without semantics.

### 3.3   Effectiveness of Semantics

We conduct an ablation study to evaluate the necessity of introducing semantics in our model. For comparison, two different training methods are used for reconstruction: reconstructing with and without semantic features. For model without semantics, we remove the semantic decoder and replace its image generator with a standard pix2pix model, which is trained for translating shapes to images directly.

As shown in Figure 5a, using the image translation models to reconstruct images only from shapes perform well on part of the samples, which are similar to the results reconstructed with semantic information. These successful cases without semantic information usually depend on effective and clear shape decoding performance. However, most of the fMRI signals' SNR is low [1] and many decoded shapes are similar. In Figure 5b, the GAN model can not deduce right decisions only from these noisy or similar shapes (left-hand side of Figure 5b), which causes the reconstructed images rendered uncorrelated details (like colors). Images reconstructed from the same shapes with semantic information are showed in the right-hand side of Figure 5b. The generated images' colors are corrected with the guidance of the semantic information. Quantitative results are showed in Figure 5c. The images reconstructed with semantics perform better than those without semantics (65.3% vs. 62.5%). The results indicate that by reconstructing with categorical information in semantic features, our model is able to improve the reconstruction performance visually and can reconstruct images more accurately.

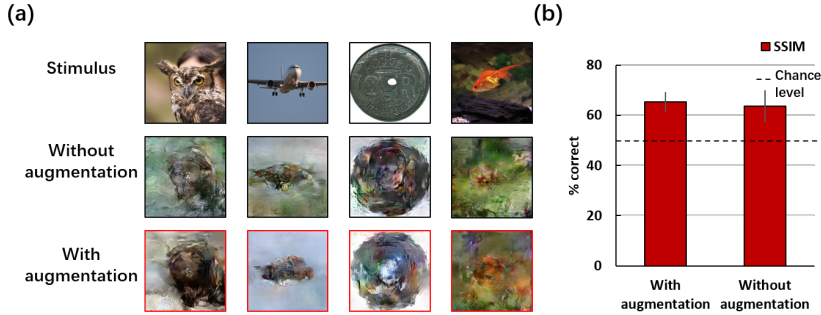

Figure 6: Effectiveness of augmentation. (a) Comparison of images reconstructed with/without augmentation. (b) Quantitative comparison of reconstruction with/without augmentation.

## 3.4 Effectiveness of Data Augmentation

To evaluate the improvement of reconstruction quality by introducing data augmentation, we train models with and without augmentation respectively, and compare the models on the test set. One model is trained on such augmented dataset and the other one is trained solely on the origin training set as a contrast. The results are showed in Figure 6. In visual comparison, the images reconstructed with augmented data look more natural and more close to the ground truth images than those reconstructed only from the origin training set (Figure 6a). In quantitative comparison, model trained with data augmentation performs slightly better than that without augmentation (65.3% vs. 63.6%). By adding more images in the GAN training phase, the short board of the limited dataset size can be complemented and the model will learn the distribution over more natural images, contributing to the improvement in reconstruction.

## 4   Conclusion

In this paper, we demonstrate the feasibility of reconstructing stimulus images from the fMRI recordings by decoding shape and semantic features separately, and merge the shape and semantic information to natural-looking images with GAN. This 'divide and conquer' strategy simplified the fMRI decoding and image reconstruction task effectively. Results show that the proposed Shape-Semantic GAN improves the reconstruction similarity and image quality.

## Broader Impact

The proposed Shape-Semantic GAN method provides a novel solution to visual reconstruction from brain activities and present a potential brain-reading technique. This method can help people recognize the human perception and thinking, and may help promote the development of neuroscience. However, the development of such brain-reading method may invade the privacy of the information within people's mind, and may cause people to worry about the freedom of thought.

## Acknowledgment

This work was partly supported by grants from the National Key Research and Development Program of China (2018YFA0701400), National Natural Science Foundation of China (61906166, U1909202, 61925603, 61673340), the Key Research and Development Program of Zhejiang Province in China (2020C03004).

## Footnotes

*Corresponding authors: Yu Qi and Gang Pan

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
