[Supplementary Material]

**Data Preprocessing** The fMRI recordings of different subjects were prepossessed using the same method and we trained models for each subject respectively using the same architecture. In our method the purpose is to transfer an input fMRI recording to the corresponding stimulus image. The fMRI data come from Shen *et al.* which contains recordings from 3 subjects. Each run's first 8s recordings are discarded. Then the voxel amplitude is averaged within each stimulus block (8s for the training set and 12s for the test set), and the fMRI recordings within each block is treated as one fMRI sample (corresponding to one stimulus image). The fMRI recordings are shifted by 4s. Before training the fMRI signals are normalized across the training/test dataset. The training images come from ImageNet (Deng *et al.*, 2009; 2011, fall release). The images of different sizes were cropped to center and were resized to images of $256 \times 256$ pixels with 3 channels.

**Details of Decoding Effects from Different ROIs** To compare the semantic decoding qualities from different ROIs' signals, we introduced a DNN classification task as the measurement. However, directly predicting the labels of stimulus images in this dataset is inappropriate for comparison because the dataset contains a total of 150 classes while have only eight different images are there in one class, which leads to problems in network training. Therefore, we merged the original 150 categories into 10 coarser categories (like mammal, birds, transportation, plant, etc). We conducted this ten-class identification task by training this DNN on 5960 fMRI samples and test on 40 reserved samples.