[Reviews · NeurIPS 2020]

Review 1

Summary and Contributions: I have read the author response, read the other reviews, participated in the discussion with the reviewers and area chair. My score remains the same. The authors propose an approach for reconstructing stimulus images used in an fMRI experiment, from the resulting imaging data. The approach follows prior work in separating the processes of decoding a representation of the imaging content and generating a plausible candidate, given that the latter allows other information besides the imaging data (e.g. priors on stimuli) to be brought in, and training of the generation model without imaging data. The approach differs from preceding ones by explicitly separating the decoding of shape and semantic representations, to be used as input for the generator. The authors demonstrate state-of-the-art reconstruction performance, relative to competing approaches.

Strengths: strengths: - generally clear paper - leverages insights from neuroscience about types of information represented in different areas to improve decoding - shows that visual and semantic information about an image can be combined effectively into a GAN (and decoded vectors are good enough for that) - comparison of decoder effectiveness for different regions of interest has neuroscience value

Weaknesses: weaknesses: - the comparison procedure with other approaches needs to be improved (details provided in Correctness) - no account taken of other ways of representing semantic information (but these would likely just improve results, which are already quite good)

Correctness: Generally, yes. 2.5) I understand the advantage of training on shape and semantic representations of stimuli to generate the reconstruction. The discussion of augmentation made me wonder what stimulus set was being used to train the GAN. Just the training images for which there is fMRI data? If so, can you confirm that the 50 images used as stimuli for the fMRI test set were left out of this process? If not, I think you would be biasing the evaluation. This needs to be clarified. 3.2) Please define structural similarity index, it is not as widely known as the correlation coefficient. Is Figure 2a) displaying results for test stimuli or training stimuli? The comparison between Pearson correlation or SSIM scores of different methods does not take into account the fact that those scores are estimates, with uncertainty depending on the size of the test set (small). Given that the goal is to compare them, it is not sufficient to do this by adding error bars to each result, and see if the resulting intervals overlap. This is because the examples on the test set are the same for all the methods being compared, rather than independent samples. Moreover, there are also multiple comparisons being made and not accounted for. If comparing only two methods, it would be sufficient to perform a paired t-test given the predictions of either method over all examples. Comparing more than two methods over the same dataset could be done by considering paired t-tests for all pairs, but this might be inefficient given the need for multiple comparisons. In that situation, the appropriate approach is a Friedman test, with post-hoc tests for comparison. This also allows a generalization of the confidence interval called a "critical difference" diagram. This is all discussed in detail "Statistical Comparisons of Classifiers over Multiple Data Sets" by Janez Demsar, a paper in JMLR 7 (2006) 1-30.

Clarity: For the most part, yes. The clarification questions I have are: 2) Across this section, what data is used to pick hyperparameter values (specific coefficients, architecture of networks, etc) or, if no data are used, what are the rationales? 129: Why use a DNN for the semantic features instead of a linear regression, as for the visual ones? Why Tanh instead of ReLU (possibly requiring more layers, but often more effective) If I have to guess, you are trying to solve the problem of how to represent the items, and how to decode them, at the same time. Implicitly, this means using a very rudimentary semantic representation of each item, given that you are only trying to distinguish 10 coarse categories. Why do this instead of using an existing representation capturing richer information (e.g. Binder's 65-dimensional embedding, or a word embedding, to name just two possibilities)? 231) "that do not use test data to maintain consistency in training phase" What does this mean? 3.6) This was confusing. The augmentation here appears to be done with completely novel stimuli, whereas before it seemed to be the case that they came from the set of stimuli for which you had fMRI data. This may be resolved when the comment pertaining to that is clarified.

Relation to Prior Work: 6. Relation to prior work: Is it clearly discussed how this work differs from previous contributions? The prior work discussion is excellent, and covers almost all relevant work I am aware of, in particular "Bayesian reconstruction of natural images from human brain activity" by Naselaris et al. 2009 which introduces the concept of a semantic prior in decoding, other papers using GANs for reconstruction, and "End-to-End Deep Image Reconstruction From Human Brain Activity" by Shen et al. 2019 improving the GAN aspect by bringing in semantic priors. This work appears to be a significant extension of the latter. The one thing I think is missing is a mention of other ways of transforming stimuli into semantic information, e.g. word embeddings for the words naming the stimuli or other feature representations.

Reproducibility: Yes

Additional Feedback:


Review 2

Summary and Contributions: This paper used a new method for separately decoding shape and semantic information about images from the corresponding fMRI responses, and then combining the shape and semantic information together in order to reconstruct the image. This method seems highly effective – much more so than previous “all-in-one” methods.

Strengths: The method developed in this paper seems highly effective, a clear improvement upon earlier methods. The additional tests done by the authors—examining the effects of combining shape and semantic information and data augmentation—seem to rigorously show why this method work so well. This is a solid contribution.

Weaknesses: I cannot identify any major weaknesses in this paper. The results are solid and seem thoroughly explored. There is no major gap or mis-step.

Correctness: No issues.

Clarity: The paper was well-organized and mostly clear. The writing is at some points slightly hard to parse, and could use some additional editing.

Relation to Prior Work: Only one concern here: the idea of combining separate shape/structural and semantic decoders to reconstruct images from fMRI data is not entirely novel, but was actually the approach taken in Naselaris et al. Neuron (2009) (citation [17] in this paper). In my opinion, it would be appropriate to re-frame the contribution of this paper as updating that basic idea, which was highly effective but seemingly ignored by many papers between 2009 and the present day, using modern methods that yield much better performance.

Reproducibility: Yes

Additional Feedback: Please don't use the word “seductive” to describe a scientific goal.


Review 3

Summary and Contributions: The paper proposes a modular method of reconstructing stimuli images from the corresponding fMRI signals. The method utilizes separate models to decode shape from lower visual cortex (LVC) signals and semantic features from higher visual cortex (HVC) signals. The shape information is extracted using three decoders of varying resolution to simulate the processing in the V1, V2, and V3 areas of the lower visual cortex. The semantic information is extracted from the penultimate layer of a deep neural network (DNN) trained to classify the HVC signals. The shape and semantic information are combined in a generative adversarial network (GAN) where the generator takes a U-Net structure. The output of the generator produces an image resembling the original stimuli image. Additionally, the paper provides studies into the benefits of data augmentation when training the GAN.

Strengths: The paper contributes a novel method for reconstructing images, which attempts to replicate the hierarchical processing within the brain. The authors adequately justify and explain their theoretical grounding with several citations demonstrating the hierarchical structure of the human vision system. The roles of the LVC and HVC are heavily cited, and the paper provides its own study (Section 3.4 Decoding Effects from Different ROIs) to show the roles translate to their model. While the results are still not at the quality for their proposed "brain-reading" application, the paper shows a significant step towards a system with such potential capabilities. The main contribution of this paper is a method for efficiently extracting features for better decoding performance.

Weaknesses: (a) The paper falls short in providing a clear theoretical backing for using the U-Net structure in the generator. The explanation in Section 2.4 seems to suggest that the U-Net structure was selected due to the benefits of its bottleneck passing low-level features, but later states that it is only passing high-level features. Clarification of this section is needed. --> Since the author already clarified the advantage of using U-Net structure, I think this weakness can be removed now. (b) The paper also lacks a theoretical justification for the necessity of the HVC signals. As the brain encodes the LVC signals into the HVC signals, the information from HVC signals should also be contained in the LVC signals. Why then, do we need the high level representation for the decoding problem? Could the semantic features not be extracted straight from the LVC signals where there is more information? The reason why HVC features are needed here seems to be the way in which you construct the LVC features: since there are only “shape” features constructed from LVC, which does not contain the color or semantic features. HVC signals are required in this case. A justification for using the LVC signals for only “shape” features and not semantic features as well would better solidify the theoretical reasoning. (c) Moreover, the semantic feature construction uses a network trained for classification. A possible problem may occur if the classification network is well learned since the network is trying to extract higher level features from the already high-level representation of the HVC, t. As features become higher level, they should become invariant to many variations within a class. For each category the deep feature vector (i.e., penultimate hidden layer) would be very similar, and hence will not be very representative of the semantic features for different objects of the same class . For example, attributes of same class objects like color may get lost. Therefore, the algorithm wouldn’t really learn to decode the image, but would rather produce a prototypical color image of the object. (d) Some necessary details are missing or unclear. For example, in the data-augmentation section, is the category-average semantic vector included in addition to the fMRI decoded semantic vector when fed into the bottleneck of the generator? If so, this should be stated clearly. Besides, the protocol for the training-testing set split, learning rate, or other technical details of the algorithm need to be explicitly stated. (e) Details regarding the preprocessing of fMRI data is missing. For instance, If one subject’s V1 region is larger than another subject’s, how do you ensure the feature vectors or fMRI signals are of the same dimension? Do you do subject level alignment as well? Better clarifications are needed for this, though there is a reference for it.

Correctness: Yes, the method and claims appear to be correct.

Clarity: The paper contains a few grammatical errors. Most commonly, the article "the" is missing. Additionally, a few passages are confusing and should be reworded for clarity. Notably, in the Contributions: "Referring to the methods of image translation to reconstruct natural-looking images with GAN and high-quality reconstructions have been obtained", in Section 2.3 Semantic Decoding: "However, directly predicting the stimulus..." and in the Broader Impact: "... may promote people's cognition of human perception and thinking, can lead to the development of neuroscience", to state a few examples. The paper should also include supplemental material describing their experimental procedure further. The reader is left to guess on the exact methodology on several parts of their pipeline. For example, it would be helpful to know how the fMRI data was preprocessed. There is also confusion on the implementation of the shape decoder. In one section, the authors state that the stimulus images are binarized, but then later say that the pixels are normalized. This procedure could be clarified with a supplemental figure.

Relation to Prior Work: Yes, the paper describes how this work takes a hierarchical approach towards reconstructing images from fMRI data. By using separate models for LVC and HVC signals, this paper is able to differentiate their work and results from previous contributions.

Reproducibility: No

Additional Feedback: 1) I would like to see an expanded or clarified explanation of why and how the Imagenet classes were reduced from 150 classes to 10 classes. --> The author replied in the rebuttal, and also said they will add more details in supplementary material. I think this can be removed now. 2) As mentioned in the Weaknesses, I would like to see some classification in the justification for the U-Net structure. You seem to suggest the U-Net structure was selected because its bottleneck structure could pass the low-level shape features with skipping connections, but later, you state the bottleneck represents the high-level features of the shape image. Do the low-level features still pass through? --> Since the author already clarified the advantage of using U-Net structure, I think this can be removed now. 3) The Broader Impact section could use a discussion on the potential negative ethical and societal implications of your work. Applications such as "mind-reading" hold a potential for harms against personal freedom and security. 4) For acronyms like "ROI", please spell them out in the first use within the paper. A more explicit explanation of your definition of "shape" would be useful. You describe them as "man-made image features", but they appear to be similar to a segmentation mask. 4) Was the testing set withheld for the training of the shape and semantic decoders as well? It is unclear whether the testing set was withheld of the training of the whole pipeline or just the GAN training. --> Since the author clarified that the testing set is strictly withheld, I think this can be removed now. 5) In equation 4, the adversarial loss term, Ladv, should be a combination of the discriminator loss and generator loss terms. The generator loss term should be explicitly stated similar to the discriminator loss. 6) In line 168, you state that you use the L1 loss, but then in equation 2, you use the L2 norm. 7) In lines 239-241, you claim to train the semantic decoder, but then later state that you pass the test fMRI data through a pretrained decoder. It should be clarified if you are training the decoder or using a pretrained decoder. 8) For equation 1, consider revising your notation. As it currently stands, it is difficult to tell if η are the weights for all the decoders or for just one decoder. Maybe consider adding a “k” subscript to the η.

[Author Response · NeurIPS 2020]

We thank all reviewers for the valuable and constructive comments.

**To Reviewer #1**

**What stimulus set was being used to train the GAN, and for augmentation?** The fMRI dataset includes 1200
training images (150 categories) and 50 test images with the corresponding fMRI recordings. The GAN was trained
with the 1200 training images in the fMRI dataset. In augmentation, we sampled extra 200 images from the ImageNet
dataset. The images used for augmentation have no overlap with both training and test images from the fMRI dataset.

**Is Figure 2(a) displaying results for test stimuli or training stimuli?** Figure 2(a) displays the results for test stimuli.
We will make it clear in the final version. Thanks.

**How did the hyperparameter values being selected?** In the training of GAN, 100 fMRI recordings of the training set
were kept as a validation set for hyperparameter selection. Then we used the whole training set (containing the samples
in validation set) to train the model with the selected hyperparameters. We will make it clear in the final version.

**Why use a DNN for the semantic features instead of a linear regression, as for the visual ones? Why Tanh**
**instead of ReLU?** We have evaluated both DNN and linear regression for semantic decoding and did not put it in
the paper for space limit. In this experiment, the DNN model obtained better performance (53.9% v.s. 49.1% in
classification accuracy). About the activation function, using Tanh function can converge stably in this experiment. We
agree that functions like ReLU can also work in this model. We will explore different models for semantic decoding in
the future study.

**About the evaluation and comparison criterion.** Thanks for your constructive suggestions. We did not conduct other
comparison methods because the codes of some papers were not available during the experiment. We are trying to
supplement some results of the Friedman test in the final version as suggested.

For the other suggestions/issues, we will revise the paper accordingly. Many thanks for your valuable comments.

**To Reviewer #2**

**About the contributions.** Thanks for confirming our contributions. The previous study [17] provides a good basis and
support for our approach. We will clarify the contributions in the introduction, and add extra discussions in the final
version. In addition, we will add necessary details to the unclear parts in the article.

**To Reviewer #4**

**Why using the U-Net structure in the generator?** The advantage of using U-Net lies in that it both processes
high-level information (passed through in the bottleneck) and preserves low-level structures (by skip connections). For
image reconstruction, we need to combine both high-level (category) and low-level (shape) features, thus U-Net is a
suitable choice.

**Regarding the necessity of the HVC signals and loss of information.** We agree that theoretically LVC information
should contain the information from HVC. However, in the process of recording fMRI signals, the LVC signals cannot
be guaranteed to contain all the useful information. Experiment 3.4 compares the performance of decoding semantic
information from different cortical areas, which shows that fMRI signals from HVC contain more reliable high-level
information (see Figure 3(a)). Therefore, the fMRI signals from HVC should be much helpful for the reconstruction.
In reconstruction, our approach did not explicitly model other features (e.g. color). However, part of the information
might be implicitly encoded in the weights of the GAN model during the training phase. In the future work, we will try
to explicitly decode more critical information for improvement.

**Why and how the images' classes were merged?** In the original dataset there are only 8 images within one category
(totally 150 categories). Training the DNN model with so few samples within one category is difficult. To facilitate
the model's training, the number of images in each category was increased by merging similar classes. We manually
grouped similar images from subdivide categories (e.g. gorilla and chimpanzee) into new coarser categories. We will
add a supplementary material to explain the details.

**Was the testing set withheld for the training of the decoders as well?** Yes. The test set was strictly withheld in the
training of the shape decoder, semantic decoder, and the GAN model. Thanks.

**About details of implementation.** We will include a supplemental material to describe the preprocessing of fMRI
data and network implementation details. And we will explain in detail what is not clear in the paper. For the other
suggestions/issues, we will revise the paper accordingly. Many thanks for your valuable comments.

[Meta-Review · NeurIPS 2020]

This paper was well received, and the reviewers praised it for its clarity and contributions. The idea to separate shape and semantics in the reconstruction is an interesting approach, and one that has proven quite useful. This work is likely of interest to a wide range of the NeurIPS community, those interested in computer vision as well as neuroscience. The reviewers pointed out a few places that could be clearer, but these points (for the most part) could be handled in minor revisions. One point remained unanswered about the loss of information in the HVC, and was discussed by the reviewers after seeing the rebuttal. One reviewer thought that section 3.4 didn't prove that the LVC signals do not contain the same necessary information as the HVC. Instead, it shows that the HVC signals perform better with their particular DNN architecture. Since the LVC signals have gone through less processing in the brain, it is possible that they simply require a larger network to extract the relevant information. Perhaps the authors could add this caveat to their final paper.